# *N*-Butyldeoxygalactonojirimycin Induces Reversible Infertility in Male CD Rats

**DOI:** 10.3390/ijms21010301

**Published:** 2019-12-31

**Authors:** Vijayalaxmi Gupta, Sheri A. Hild, Sudhakar R. Jakkaraj, Erick J. Carlson, Henry L. Wong, C. Leigh Allen, Gunda I. Georg, Joseph S. Tash

**Affiliations:** 1Department of Molecular and Integrative Physiology, University of Kansas Medical Center, Kansas City, KS 66160, USA; 2Division of Reproductive Endocrinology and Toxicology, BIOQUAL, Inc., Rockville, MD 20850, USA; sheri.hild@nih.gov; 3Department of Medicinal Chemistry, Institute for Therapeutics Discovery and Development, College of Pharmacy, University of Minnesota, Minneapolis, MN 55414, USA; jakka002@umn.edu (S.R.J.); carl2810@umn.edu (E.J.C.); hlwong@umn.edu (H.L.W.); clallen@umn.edu (C.L.A.); georg@umn.edu (G.I.G.)

**Keywords:** iminosugar, contraceptive, NB-DGJ, NB-DNJ

## Abstract

This study shows for the first time that an iminosugar exerts anti-spermiogenic effect, inducing reversible infertility in a species that is not related to C57BL/6 male mice. In CD rats, *N*-butyldeoxygalactonojirimycin (*N*B-DGJ) caused reversible infertility at 150 mg/kg/day when administered daily as single oral dose. *N*B-DGJ inhibited CD rat-derived testicular β-glucosidase 2 (GBA2) activity at 10 µM but did not inhibit CD rat-derived testicular ceramide-specific glucosyltransferase (CGT) at doses up to 1000 µM. Pharmacokinetic studies revealed that sufficient plasma levels of *N*B-DGJ (50 µM) were achieved to inhibit the enzyme. Fertility was blocked after 35 days of treatment and reversed one week after termination of treatment. The rapid return of fertility indicates that the major effect of *N*B-DGJ may be epididymal rather than testicular. Collectively, our in vitro and in vivo studies in rats suggest that iminosugars should continue to be pursued as potential lead compounds for development of oral, non-hormonal male contraceptives. The study also adds evidence that GBA2, and not CGT, is the major target for the contraceptive effect of iminosugars.

## 1. Introduction

Access to orally bioavailable, safe, effective, and reversible male contraceptive agents would be an important advance in expanding current contraceptive options available to couples. While female oral hormonal contraceptive agents have been on the market since 1960, the development of male hormonal contraceptives has been slow [1,2,3]. Since the 1950s, a multitude of clinical trials have been conducted to develop hormonal male contraceptive agents, but side effects and pharmacokinetic issues have made it difficult to bring such agents to market [4]. Non-hormonal experimental approaches for male contraception using gossypol [5] and alpha-chlorohydrin [6] showed promise in animals and in early stage human studies, but neither compound has proven safe or effective enough to be acceptable for human use [7]. Therefore, the discovery of alternative lead compounds is important for the development of male non-hormonal contraceptive agents [8,9]. Glycosphingolipids are among possible targets for non-hormonal contraception. They are abundant in sperm [10,11] and thought to be necessary for normal spermatogenesis and spermiogenesis. Mice deficient in enzymes involved in glycolipid synthesis, such as ceramide-specific glucosyltransferase (CGT) and β-glucosidase 2 (GBA2) are known to have severely impaired fertility [12,13,14,15,16]. CGT is the enzyme that catalyzes the first step of glycolipid biosynthesis, the reaction between uridine diphosphate glucose (UDP-Glu) and ceramide (Cer) to form glucosylceramide (Figure 1), a process that eventually leads to acrosome formation. GBA2 has the opposite function and catalyzes the cleavage of glucosylceramide into glucose (Glu) and ceramide. GBA2-deficient mice have increased levels of glycosylceramide in testes, liver and brain [17].

Iminosugars, which are sugar analogs carrying a nitrogen at the position of the endocyclic oxygen atom, are one of the most interesting classes of glycomimetics [18]. The *N*-alkylated iminosugar *N*-butyldeoxynojirimycin (*N*B-DNJ, Figure 2) inhibits somatic and testicular CGT (ceramide-specific glucosyltransferase) in vitro and in vivo [19,20,21,22] and was shown to be an effective and reversible oral male contraceptive agent when tested in C57BL/6 mice. Reversible inhibition of fertility was observed three weeks after daily oral administration of 15 mg/kg *N*B-DNJ [22]. Fertility returned three weeks after cessation of *N*B-DNJ administration [22]. Spermatozoa obtained from the treated mice exhibited abnormal acrosomes, disorganized tails, small and widely spaced mitochondrial sheaths and nuclear dysmorphia [22]. Testicular histology revealed that iminosugar *N*-butyldeoxygalactonojirimycin (*N*B-DGJ, Figure 2) also disrupts spermiogenesis at a dose equivalent to 1200 mg/kg of *N*B-DNJ [22]. *N*B-DGJ is known to be a more selective inhibitor of glycolipid biosynthesis than *N*B-DNJ and to have fewer adverse side effects in mice, especially at high dosage (2400 mg/kg) [23].

Subsequent studies revealed that the activity of *N*B-DNJ was species specific and only effective as a male contraceptive in C57BL/6 and C57BL/6-related mice, but not in other mouse strains tested, nor in the Russenkaninchen rabbit strain (Crl:CHBB (HM)) [24]. *N*B-DNJ also had no effect in human subjects, and sperm counts, and semen parameters remained normal [25,26,27]. *N*B-DNJ (USAN name, miglustat) is a clinically utilized agent for the treatment of type 1 Gaucher disease, a glycosphingolipid storage disease [18].

Later, iminosugars *N*B-DNJ and *N*B-DGJ were found to inhibit GBA2 at lower concentrations than CGT, and the levels of glucosylceramide were increased in the testes, brain, and spleen of mice treated with iminosugars [15,21]. Thus, the inhibition of GBA2 appears to be the more likely mechanism of action of the iminosugars. Recent studies showed that in GBA2 knock-out mice, glycosylceramide accumulation results in cytoskeletal defects in Sertoli cells and germs cells. In particular, F-actin structures in the ectoplasmic specialization as well as microtubule structures in sperm were affected as early as the first spermatogenic wave [16]. Effects on F-actin structures in the ectoplasmic specialization were also observed in animals treated with the male contraceptive agents adjudin and gamendazole [28,29,30].

Interestingly, the level of glucosylceramide in the testes was also increased in mouse strains that remained fertile after iminosugar treatment, indicating that the sensitivity to glucosylceramide may be species dependent or that additional mechanisms may be at work [21]. Since the iminosugars do not seem to interfere with meiosis and therefore do not affect spermatogonial cells and spermatocyte populations, their effects should generally be reversible. These properties make iminosugars attractive lead compounds for the development of new analogs that possess contraceptive efficacy across a broader range of species, including humans.

To further investigate the contraceptive properties of iminosugars, we designed a study to (1) determine whether *N*B-DGJ shows anti-spermiogenic efficacy in CD rats when administered as a daily single oral dose over 35 consecutive days, (2) compare the relative inhibitory efficacy of *N*B-DGJ against testis CGT and testis GBA2 from C57BL/6J mice and CD rats and also Long Evans (LE) rat testicular microsome preparations to determine strain specificity, (3) compare the in vivo anti-spermiogenic efficacy in rats with the in vitro biochemical efficacy in mice and rats to determine whether the results obtained in the in vitro enzyme assays correlate with observations made in the in vivo studies, and (4) to perform a pharmacokinetic study with *N*B-DGJ to determine if plasma levels of *N*B-DGJ in CD rats dosed orally with a single bolus were sufficient to inhibit GBA2 activity.

## 2. Results

### 2.1. Synthesis of Iminosugars

The synthesis of *N*-butyl-1-deoxynojirimycin (*N*B-DNJ, Figure 2) and *N*-butyl-1-deoxygalactonojirimycin (*N*B-DGJ, Figure 2) was accomplished following known procedures (detailed in Appendix A) using 2,3,4,6-tetra-*O*-benzyl-α-glucopyranose and 2,3,4,6-tetra-*O*-benzyl-d-galactopyranose, respectively, as the starting materials [31,32]. Since the iminosugars are likely to be administered orally to men in a single dose, we selected a once a day oral drug dosing regimen for our study. This protocol would also avoid potential problems from food digestion that could influence drug bioavailability and metabolism.

### 2.2. NB-DGJ Impact Testis Histology and Sperm Acrosome in C57Bl6J Mice

It is known that NB-DNJ caused infertility in male C57BL6J mice [22], so we carried out in vivo studies using NB-DNJ as positive control to test whether NB-DGJ affected testicular histology and sperm morphology. Mice were treated with 15, 15 or 50 mg/kg NB-DNJ or NB-DGJ or water (vehicle). While testis obtained from vehicle group showed normal spermatid with elongated head, both NB-DNJ and NB-DGJ treated mice testis had rounded, abnormally shaped spermatid heads (Figure 3a).

Sperm acrosome morphology analyzed by using FITC-PSA staining showed that NB-DNJ at all three concentrations caused abnormal acrosome in all spermatozoa, whereas NB-DGJ caused a dose-dependent increase in the number of abnormal spermatozoa, as indicated in Table 1, Figure 3b.

Our observation and published reports encouraged us to test NB-DGJ in higher rodent species, so we used male CD rats to carry out fertility studies.

### 2.3. NB-DGJ Induced Reversible Infertility in Male CD Rats

It had been demonstrated [33] that 150 mg/kg *N*B-DGJ caused reversible infertility in C57BL/6J male mice, so we decided to use the same dose in rats assuming that this regimen would provide a relatively higher drug exposure in rats than in mice. This rationale is based on the fact that the dosage of a compound when tested in a higher species depends on blood volume and body surface area rather than body weight. We conducted mating trials with two groups of six male CD rats each that were treated for 35 consecutive days (weeks 0–4; total 5 weeks) with an oral daily dose of 150 mg/kg *N*B-DGJ, or with vehicle control (water), as detailed in the Experimental Section. Each male rat was cohabited with two female rats weekly during weeks 1 through 9 of the trial. The females were necropsied seven days after separation from the males to determine the number of conceptuses. The male rats were necropsied at week 11 (Appendix A).

The fertility of the male rats was determined by counting the number of normal and abnormal conceptuses in pregnant female rats. The rat mating trial with *N*B-DGJ at a dose of 150 mg/kg/day induced reversible infertility (Table 2 and Figure 4). Female rats mated with vehicle treated male rats showed an average of 14 normal conceptuses per pregnant female throughout the entire study. In the treatment group, the female rats mated with *N*B-DGJ treated male rats produced the same number of normal conceptuses as females mated with vehicle treated rats in the first week of mating. However, in the second week, the numbers of normal conceptuses were reduced by 50% as seen in Table 2. The male rats were still designated as fertile, as they were able to impregnate the female rats. At week three, 50% of the NB-DGJ treated rats were rendered infertile, and by week four, 100% of the NB-DGJ treated rats were infertile. Fertility returned rapidly, within one week (week 5, Table 2, Figure 4), following termination of the dosage regimen as opposed to the gradual loss of fertility seen during the initial three weeks (weeks 2–4, Table 2, Figure 4) of the *N*B-DGJ mating studies.

### 2.4. NB-DGJ Significantly Reduced Epididymal Sperm Motility and Caused Altered Morphology

The effects of *N*B-DGJ on rat body weight, testis weight, prostate weight, sperm count, sperm motility, and serum hormone values were examined. At week 11 (completion) of the study, the male rats were weighed, anesthetized, and exsanguinated. No significant difference (Student’s *t*-test) was observed between vehicle and *N*B-DGJ treated groups for final body weight (*p* < 0.673), paired testis weight (*p* < 0.167), spermatid head count (*p* < 0.254), paired epididymal weight (*p* < 0.184), seminal vesicle weight (*p* < 0.233), or ventral prostate weight (*p* < 0.254) (Table 3). However, epididymal sperm motility determined at week 11 (* *p* < 0.05) as per Student’s *t*-test was significantly lower in *N*B-DGJ treated rats than those treated with the vehicle (60% in *N*B-DGJ treated rats compared to 87% in control rats, Table 3; Appendix A). This is in line with lower number of epididymal sperm with normal morphology in NB-DGJ treated rats compared to the vehicle group (Table 3; Appendix A).

These low values, however, did not hinder the ability of the male rats to impregnate the female rats once iminosugar administration was stopped, as seen in Figure 4. Thus, this is the first reported effect of *N*B-DGJ on fertility in rats. Whether there are strain-selective issues related to efficacy like those reported for the mice will require additional detailed dose finding and strain comparison studies.

The male rats were bled from the tail vein at weeks 0, 1, 4 and 7 and the sera were analyzed for levels of inhibin B, testosterone, and rFSH. No significant differences between treated versus control group (Appendix A) were observed (based on ANOVA). This clearly indicated that spermatogenesis was intact in the NB-DGJ treated rats and that NB-DGJ does not impact the testes, further strengthening our finding that epididymes is the target.

### 2.5. Mechanism of Anti-Fertility Effect on NB-DGJ

Our next objective was to delineate the mechanism by which *N*B-DGJ exerts its anti-fertility effect in male rats. To meet this goal, we conducted In Vitro assays to determine the potency and selectivity of *N*B-DGJ to inhibit testicular CGT and testicular GBA2. Microsomes were isolated from the testes of adult C57BL/6J male mice and CD rats as well as LE rats and used to measure the in vitro inhibition of testicular CGT by the iminosugars. The same microsomal preparation was employed to assess the in vitro inhibition of testicular GBA2. The results are shown in Table 4. The IC_50_ values were 76 and 42 μM for mouse- and LE rat-derived CGT inhibition, respectively, whereas no enzyme inhibition was observed for CD rat-derived enzyme up to 1000 μM. Testicular GBA2 inhibition by *N*B-DGJ was similar for mice and both rat strains, with IC_50_ values of ~16 μM for mice, ~14 μM for LE rats and ~11 μM for CD rats.

### 2.6. NB-DGJ Is Orally Bio-Available in CD Rats

To correlate in vivo activity with the bioavailability of *N*B-DGJ, we performed a pharmacokinetic (PK) study to examine plasma levels following a single bolus oral dose at 150 mg/kg. Briefly, male CD rats (225–235 g) were orally dosed with *N*B-DGJ by gavage (*n* = 4). At various time points following dosing, the rats were bled, and the resulting plasma was analyzed by LC/MS/MS. As seen in Table 5 and Figure 5, oral dosing of *N*B-DGJ resulted in a t_½_ of 6.5 h with a C_max_ of 11,016.7 ng/mL (~50 µM). These data are very similar to results presented in a previous study [34] that examined the pharmacokinetics of *N*B-DNJ (miglustat) using radioactively labeled compound. Nevertheless, it appears that oral dosing of *N*B-DGJ results in sufficient plasma levels to potently inhibit rat GBA2 activity in CD rats.

## 3. Discussion

Administration of *N*B-DGJ as a daily single oral dose of 150 mg/kg for 35 days caused reversible infertility in male CD rats. Epididymal sperm motility in treated males was significantly lower than in the control group. Fertility returned within one week after the dosing was stopped, indicating that *N*B-DGJ may exert its major effect on the epididymis rather than the testis, since the response occurs in less than the epididymal transit time for sperm from the rete testis. It is known that the epididymal transit time of sperm in rats is ~8 days, thus the reversal of fertility in NB-DGJ treated male rats in one week suggests that epididymal sperm were affected, leaving the testis functionally intact. Infertility in these rats lasted only till the next wave of sperm were received in the caput to undergo the maturation process. Moreover, as indicated in Table 3, mean testis weight and testicular spermatid head count were similar between the vehicle and NB-DGJ treated group.

It should be noted here that *N*B-DGJ at the same concentrations was effective in inducing infertility in C57BL/6J male mice [22]. In C57BL/6J, it took approximately three weeks for fertility to resume, which indicates that the major effect in mice may be at the testicular level [35].

In the studies reported here, the body weight and gonadal weight of *N*B-DGJ treated animals were similar to the control group animals. Circulating levels of inhibin B, testosterone, and rFSH in *N*B-DGJ treated rats showed no significant difference when compared to the control group.

We next studied the inhibition of testicular CGT and testicular GBA2 activity, derived from C57BL/6J mice, CD rats, and LE rats (Table 4). For the mouse-derived enzymes, we found that *N*B-DGJ is a weak inhibitor of CGT and that GBA2 activity is inhibited more strongly than CGT activity. Very similar enzyme inhibition trends have been reported previously [15,20,21,36] and are shown in Table 5. In one of the studies, *N*B-DGJ inhibited GBA2 with an IC_50_ of 300 nM [36], which is approximately 44 times more potent than what we found. However, the enzyme sources for the experiments in Table 6 were mostly non-testicular, not differentiated by species, or generated from transfected cells in some instances. The most potent iminosugar reported to date is an analog of DGJ that carries an *N*-(5-(adaman-1-ylmethoxy) pentyl) group and inhibits GBA2 with an IC_50_ value of 0.7 nM [15]. However, that compound has not yet been evaluated in vivo for male contraception. The corresponding DNJ *N*-(5-(adaman-1-ylmethoxy-pentyl) analog was shown to potently control hyperglycemia in a rodent model of diabetes [36].

When we investigated LE-derived CGT, we found that *N*B-DGJ is a weak inhibitor of this enzyme and that *N*B-DGJ did not inhibit CD-derived CGT up to 1000 µM. This indicates that there might be species- and tissue-dependent inhibition of the CGT enzymes.

We noted that *N*B-DGJ inhibited mouse-derived and rat-derived GBA2 at similar concentrations. This suggested that *N*B-DGJ given at relatively higher doses could be an effective contraceptive agent in rats if GBA2 is a major target for inducing sterility, which has now been confirmed several times. This was further supported by the observation that *N*B-DGJ inhibited CD rat GBA2 at 10 µM. Thus, comparing our in vitro enzyme assays using C57BL/6J mice, LE and CD rats as well as in vivo experiments performed with CD rats, it appears certain that the inhibition of testicular GBA2 is crucial to rendering male rats infertile. In conclusion, *N*B-DGJ inhibited mouse testicular and LE- as well as CD-derived rat testicular GBA2 at similar concentrations (15.7, 13.8, and 10 µM respectively; Table 3) and at a daily dose of 150 mg/kg, rendered CD rats infertile after 35 days of treatment. All evidence points towards GBA2 as a likely target for developing iminosugars as male contraceptive, as suggested previously [15]. One of the encouraging observations is the lack of adverse effects in any of the treated animals. As additional confirmation, we performed a PK study in CD rats to determine if *N*B-DGJ was sufficiently bioavailable to exert an inhibitory effect in vivo. As it is difficult to measure drug concentrations at the target organ (especially in human patients), the plasma drug concentration is typically used as a surrogate since there is typically a direct correlation between drug levels in the blood and drug levels in the surrounding tissues. The resulting analysis (Figure 4) revealed a t_½_ of 6.5 h with a C_max_ and t_max_ of 11,016.7 ng/mL (~50 µM) and 1.3 h, respectively. These data are similar to published work by Treiber, et al. for *N*B-DNJ in which they orally dosed male and female CD rats with [^3^H]-*N*B-DNJ (160 mg/kg) and reported a t_½_ = 4.5 h, C_max_ = 13,800 ng/mL (~63 µM) and t_max_ = 0.5 h [34].

Our study is the first to demonstrate that an iminosugar can induce reversible infertility in a species other than C57BL/6-related mice. The observed rapid reversibility of infertility in CD rats after cessation of drug treatment (one week), compared to mice (three weeks) suggests that other targets and/or species differences may be influencing the effects of the iminosugars. Thus, the results of our study provide new evidence that it might be possible to develop iminosugars as male contraceptive agents for human use. Our current study has limitations; however, the results are encouraging and lay a strong foundation for future investigations with higher dosage and/or modified analogs of iminosugars. We have already synthesized and determined the in vitro efficacy of various four- and eight-membered analogs of iminosugars as part of our structure–activity-relationship (SAR) approach to determine the best structure that can exhibit anti-fertility effects over a broad range of species [37]. These iminosugars are attractive lead compounds for development of non-hormonal male contraceptives, because they are orally bioavailable, are well tolerated, have no toxic side effects, and do not affect male endocrinology.

## 4. Materials and Methods

### 4.1. General Methods

All reagents, unless otherwise specified, were purchased from Sigma and were ACS grade. Thin layer chromatography (TLC) plates (silica Gel 60 A, 20 × 20 cm, layer thickness 250 μm) were obtained from Whatman International Limited, England. Bouin’s Fixative was obtained from Fisher Diagnostics.

### 4.2. Ethics Statement

For all studies using animals, the minimum number of animals to achieve statistical robustness and reduction requirements of the IACUC-approved studies (University of Minnesota Protocol# 1503-32441A, Kansas University Medical Center Protocol #s 2009-1800 and 2006-1564, and BIOQUAL Protocol #s 107 AC and 107 AD) were used. Animals for mating studies were euthanized (CO_2_ chamber) following the blood collection at 48 h in accordance with the recommendations of the American Veterinary Medical Association (AVMA) Guidelines for the Euthanasia of Animals. Animals used for harvesting testes tissues for microsome preparation were euthanized by CO_2_ asphyxiation followed by cervical dislocation, as per IACUC protocol approved by University of Kansas Medical Center.

### 4.3. Effect of NB-DGJ on C57BL6J Mice Testis and Spermatozoa

Groups of six mice, each, were used for three different doses of NB-DNJ or NB-DGJ formulated in milliQ water at 50 mg/kg, 25 mg/kg and 15 mg/kg, respectively. Control groups were given an equivalent volume of water. Animals were given compound or control once daily for 35 consecutive days via oral gavage using animal feeding needles (Biomedical Needles, Popper, NY, USA). Body weight was noted every week for all treatment groups. Mice were euthanized at the end of a 35 days dosing regimen using CO_2_ asphyxia followed by cervical dislocation. Testes were harvested, weighed and stored in Bouin’s fixative for 48 h, then sliced into two halves, and stored in fresh Bouin’s for another 24 h. The tissue slices were then rinsed in 70% ethanol and processed and stained for histology using a standard protocol for Hematoxylin and eosin staining. Spermatids with elongated and abnormally shaped heads were counted per 20 fields and average number calculated for 6 mice/group.

For sperm retrieval, epididymis was removed from the animal and placed in Petri dish. One mL of capacitation buffer (Capacitation buffer consisted of 110 mM NaCl, 2.68 mM KCl, 0.36 mM NaH2PO4, 25 mM NaHCO3, 0.49 mM MgCl2, 2.40 mM CaCl2, 25 mM HEPES, 5.50 mM Glucose, 1 mM Pyruvic acid, pH 7.4; Tash and Bracho, 1998) at room temperature was added to the dish. Using forceps, the caput of the epididymis was held in place and the cauda sliced repeatedly with a razor blade. The spermatozoa were allowed to disperse for 15 min with gentle swirling intermittently. A 100 µL aliquot of the sperm suspension was dropped onto a standard glass microscope slide (Fisherbrand superfrost microscope slides 25 × 75 × 1.0 mm)) and smeared with side of pipette tip. The slides were air dried flat for 1 h at room temperature. The sperm smear was fixed in methanol for 2 min. The sperm smear was than stained using FITC-PSA (Fluorescein isothiocyanate Pisum Sativum Agglutinin; Vector Laboratories, Burlingame, CA) was diluted 1:7000 in PBS containing 1:10,000 DAPI dihydrochloride, FluoroPure™ grade, Invitrogen) (≥98% pure) to make the stock solution. To each slide, 100 µL of FITC-PSA was added and spread out with edge of pipette tip to cover the dried sperm smear. The slides were incubated at 37 °C for 10 min, and then washed with 5 changes of PBS for 5 min each. The slides were air-dried and mounted with 20 µL Vectashield mounting medium for fluorescence (Vector Laboratories, Burlingame, CA, USA) and observed. Slides were viewed on a Nikon IX-81 microscope under a 40× objective using FITC and DAPI filter to visualize acrosome morphology. The phase contrast images were also collected at the same magnification to compare morphological details.

### 4.4. Rat Mating Trials Challenged with NB-DGJ

Eighteen adult male CD rats were divided into three groups of six rats each. One group of six CD rats was treated with vehicle (distilled water containing 0.9% benzyl alcohol) via oral gavage (5 mL/kg/day), and the second group (6 rats) was treated with *N*B-DGJ (in distilled water containing 0.9% benzyl alcohol) via oral gavage at 150 mg/kg/day (in approximately 5 mL/kg of water) daily for 35 consecutive days. The third group (6 rats) was treated with 15 mg/kg NB-DNJ in distilled water containing 0.9% benzyl alcohol) via oral gavage (in approximately 5 mL/kg of water) daily for 35 consecutive days (Appendix A).

Rats were weighed on day 0 (first day of dosing) and weekly thereafter. Dosages were adjusted based on the most recent body weight. Males were cohabited with untreated females (one male per two females) weekly at weeks 1 through 9 [38,39]. Females were necropsied seven days after separation from males to determine the numbers of implantation sites. Individual male rats were bled from the tail vein at weeks 0, 1, 4, and 7 and serum was harvested by centrifugation. At week 11, the male rats were weighed, anesthetized, and exsanguinated. Sera were analyzed for inhibin B by ELISA (enzyme-linked immunosorbent assay), and rat follicle stimulating hormone (rFSH) and testosterone by radioimmunoassay [38]. Testes, epididymides, ventral prostates, and seminal vesicles were excised and weighed. The left testis was homogenized, and spermatid head counts obtained [39].

Sperm were obtained from the left cauda epididymis and evaluated for sperm motility, as follows. The cauda epididymis was punctured with an 18-gauge needle and sperm expressed into the well of a culture slide. The sperm were diluted with capacitation buffer [40] containing 0.1% BSA to obtain about 20–40 sperm per field as viewed in the inverted microscope. The number of motile and non-motile sperm was determined, and the percent of motile sperm were calculated. A smear of the diluted sperm was also prepared and used to assess sperm morphology under phase contrast. The percent of normal/abnormal sperm were calculated.

Statistical analysis using the Student’s *t*-test was performed on body weights, paired testes weights, ventral prostrate weights, seminal vesicle weights, paired epididymal weights, spermatid head counts, percent motile and normal epididymal sperm. We used analysis of variance (ANOVA) for serum inhibin B, rFSH, and testosterone, since these measurements were repeated throughout the study period. The effect of *N*B-DNJ at 15 mg/kg/day dose in the rat model was studied following the same protocol as that for *N*B-DGJ. Since the mating trials did not show effect on fertility by week 6, the study was terminated at week 7 (data not presented).

### 4.5. Pharmacokinetic Analysis

Four male CD rats (225–235 g) were purchased from Harlan (Indianapolis, IN, USA) with in-dwelling jugular cannulas and allowed to acclimate for 72 h in climate-regulated holding rooms in the University of Minnesota’s AAALAC-certified animal facilities. Standard rat chow and water were provided ad libitum throughout the study. *N*B-DGJ was formulated as described above to 33.75 mg/mL for in vivo administration. Rats were dosed orally by using a 15 G × 3” disposable gavage needle with a maximum volume of 5 mL/kg body weight. At various time points (0.083, 0.25, 0.5, 1, 2, 4, 8, 12, 24 and 48 h) following dosing, approximately 0.1 mL of blood was collected from each rat into BD Microtainer^®^ with Lithium Heparin (Becton-Dickinson). Plasma was then isolated by allowing the samples to sit at room temperature for 30 min followed by microcentrifugation at 15,000× *g* for 5 min. Plasma samples were then stored at −70 °C until processing and analysis.

#### 4.5.1. Sample Processing

Frozen plasma samples were thawed on ice and processed by adding 2 volumes of 100% acetonitrile (containing 750 ng/mL miglitol (*N*-hydroxyethyl-1-deoxynojirimycin) as an internal standard) and vortexed. The samples were then incubated on ice for 1 h and then microcentrifuged at 15,000× *g* for 5 min. The cleared supernatants were then transferred to 12 × 32 mm glass vials with inserts (Thermo Scientific™ National™ Inserts for Target™ LoVial™ Wide-Opening Vials).

#### 4.5.2. LC/MS/MS Quantitation

Processed samples were analyzed by LC/MS/MS using a previously published method [41]. A standard curve consisting of normal rat plasma containing varying concentrations (1500, 500, 166.7, 55.6, 18.5, 6.2, 2.1, 0.7 and 0.22 ng/mL) of *N*B-DGJ was constructed and processed as described above. The analytical system was composed of a Quattro Ultima triple quadrupole MS/MS coupled with a Waters Acquity UPLC (Ultra Performance Liquid Chromatography). UPLC isocratic separation of analytes was performed using a Waters Atlantis Hilic 3 µm, 150 × 2.1 mm analytical column (Waters Corporation, Milford, MA, USA) maintained at 30 °C. The eluent was a mixture of acetonitrile/water/buffer (75/10/15, *v*/*v*/*v*) delivered at 0.25 mL/min. The buffer consisted of 100 mM ammonium acetate, pH 5. For the quantification of *N*B-DGJ and miglitol (internal standard), the transitions *m/z* 220 (Q1) → *m/z* 158 (Q3) and *m/z* 208 (Q1) → *m/z* 146 (Q3) were monitored, respectively. Data reduction was performed with QuanLynx software. PK analysis was performed by WinNonLin using a non-compartmental model.

### 4.6. Microsome Preparation from Mouse and Rat Testes

Testes in 5 g batches were placed in a 50 mL culture tube containing 25 mL of Reagent B (50 mM Tris-HCl, pH 7.4, 0.25 M sucrose and 2.5 mL of reagent A), where reagent A is 1 μg/mL antipain, 1 μg/mL leupeptin, 10 μg/mL aprotinin, 13.2 μg/mL 4-amidinophenyl-methanesulfonyl fluoride hydrochloride (APMSF) and 250 mM KCl. The testes were minced with scissors and then blended by 10 s bursts repeated 2−3 times on Power Gen 700 (Fisher Scientific, Hampton, NH, USA) at setting 5–6, while on ice. The homogenate was centrifuged at 7500 rpm for 10 min at 4 °C using a Beckman SW28 rotor (5660 g). The resulting supernatant was collected and centrifuged at 23,500 rpm for 1 h at 4 °C in a Beckman SW40 rotor. The supernatant was discarded and the pellet containing the microsomes was suspended in 600 μL of reagent D (66% (*v*/*v*) reagent C, 10 mM DTT, 8 mM EDTA, 1 mM UDP-Glu and 1% CHAPSO) and dispersed by passage through a 25-gauge needle followed by an insulin needle. (15 mL of reagent C contained 0.17% *N*-laurosarcosine, 75 mM HEPES, pH 7.4, 30% glycerol, 0.03% NaN_3_, 1.5 mM EDTA, 2.25 mL of reagent A, and 3 mM DTT.) The microsome suspension was stored as 100 μL aliquots in microcentrifuge tubes, flash frozen in liquid nitrogen for 1−2 min, stored at −80 °C, and used as needed.

#### 4.6.1. Ceramide-Specific Glucosyltransferase Assay

The following solutions were added to each assay tube (Fisher 15 × 85 mm borosilicate glass): 295 μL of assay mix (50 mM HEPES, pH 7.4, 10% (*v*/*v*) reagent A, 5 mM MnCl_2_, 0.1 mM phosphatidylcholine, 50 μM conduritol B epoxide, 2 mM EDTA, 5 mM UDP-Glu), 145 μL water, 50 μL iminosugar from serially diluted stock solutions prepared in water, and 100 μg testicular microsomes. Control tubes contained the same components, except microsomes. Reactions were initiated by the addition of 3 μL of bovine serum albumin–ceramide and incubated at 37 °C for 30 min, then terminated by addition of 1 mL of 2:1 (*v*:*v*) chloroform: methanol, vortexed, and incubated at room temperature for 30–60 min to allow phase separation. The upper phase and the mid-layer were removed and discarded, and 500 µL of chloroform:methanol: water (3:48:47) was added to the bottom layer, vortexed, and allowed to sit for 15 min at room temperature. The resulting upper phase was again removed and 100 µL of chloroform:methanol (2:1) was added, and then sample tubes were dried in a vortex evaporator overnight.

##### Thin Layer Chromatography

TLC plates (Whatman silica gel 60 A, 20 × 20 cm, layer thickness 250 μm) were pre-treated by immersion in chloroform:methanol: water (50:50:15) for 5 min, air dried for 10 min, then immersed in 5% sodium borate (prepared in methanol) for 1 min, dried and heated at 120 °C for 1.5 h. The dried sample tubes were reconstituted with 100 µL chloroform:methanol (2:1) and vortexing, and 20 µL was then spotted onto the plates at the origin. The spotted plates were air-dried and placed in a sealed TLC chamber saturated with chloroform:methanol: water (60:30:5) and run approximately 1 h until the solvent reached <1 cm from the top of plate).

##### Detection and Quantitation of Substrate/Product

The TLC plates were documented using a UV transilluminator (at 302 nm) and analyzed using AlphaEase (Fluorchem SP, Windows v5.0.2) software. The integrated density value (IDV) was plotted against iminosugar concentration using Sigma Plot 10. A linear regression plot was used to determine IC_50_ values.

#### 4.6.2. Testicular Glucosidase Assay

The assay was carried out in 96-well plates: 50 μL of 3 mg/mL 4-methyl umbelliferyl-beta-d-glucoside (MUG; Sigma, St. Louis, MO, USA) was added to each well using a multi-channel pipette, followed by 10 μL of iminosugar serial dilutions to yield 0, 5, 10, 50, 100, 500 and 1000 μM final concentrations, added from left to right, to provide increasing concentrations of the iminosugar from the top to the bottom of the plate. Then, 50 μL of testicular microsomes (1 μg/μL) was added to each well using a multi-channel pipette in the first column. Another multi-channel pipette was kept pre-loaded with terminator solution (100 μL 1 M sodium carbonate, pH 10.7) and was added simultaneously to the first row as a 0-time point. Microsomes were then added to the remaining rows after setting the timer to 1 min. Every 1 min, the terminator solution was added to each row until the 12th row. Absorbance was then detected at 360/460 nm using Synergy HT Multi-Mode Microplate Reader (BioTek). The absorbance values were subtracted from background (MUG only). A linear regression plot was created using Prism software (Graph Pad Prism 5) to determine the IC_50_ values.

## 5. Conclusions

NB-DGJ, administered orally, induced reversible infertility in male CD rats. Thus, unlike, previously reported, iminosugars exert anti-fertility effects in rodent species other than C57Bl6J, indicating that iminosugars and their derivatives should continue to be explored as potential non-hormonal male contraceptive compounds. They are attractive candidates due to their safety, efficacy and excellent bioavailability.

## Figures and Tables

**Figure 1 ijms-21-00301-f001:**
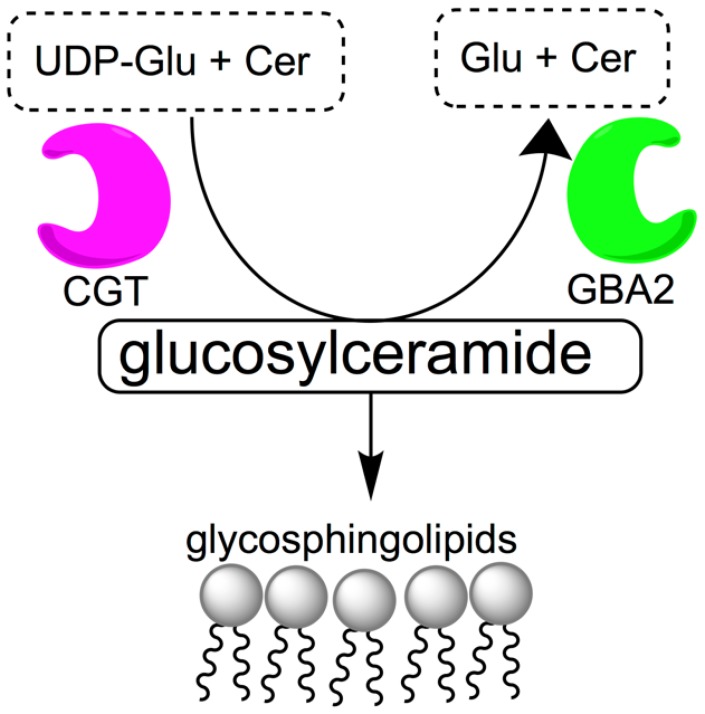
Metabolism of glucosylceramide. Ceramide-specific glucosyltransferase (CGT) is responsible for catalyzing the reaction between uridine diphosphate glucose (UDP-Glu) and ceramide (Cer) to form glucosylceramide, which is catabolized into glucose (Glu) and ceramide by β-glucosidase 2 (GBA2).

**Figure 2 ijms-21-00301-f002:**
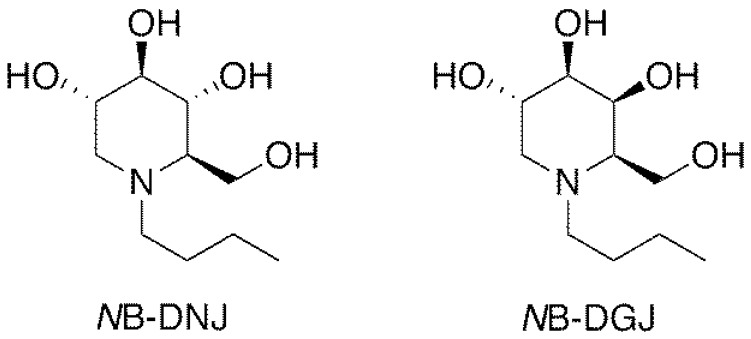
Structures of iminosugars *N*B-DNJ and *N*B-DGJ.

**Figure 3 ijms-21-00301-f003:**
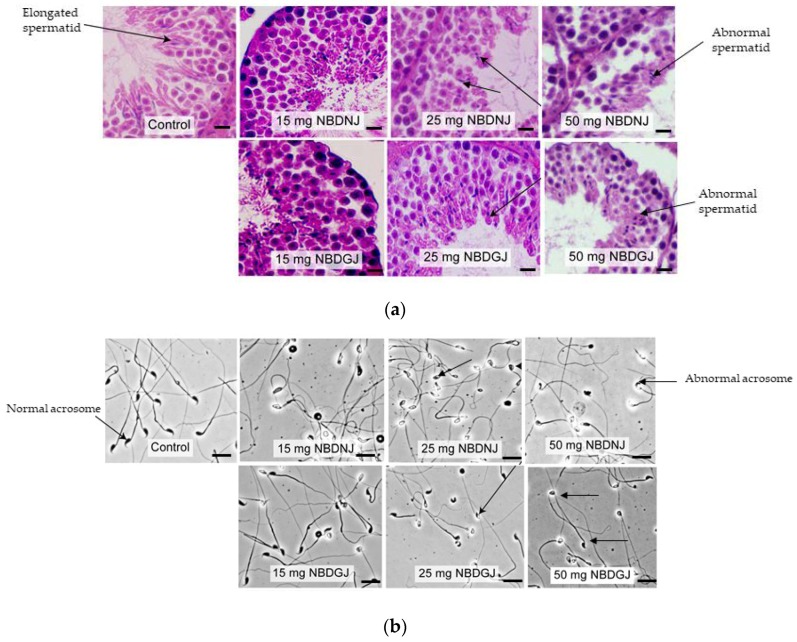
Testis histology and epididymal sperm morphology C57BL6 mice treated with vehicle and/or various dosage (15,25 and 50 mg/kg) of NB-DGJ/NB-DNJ (**a**) NB-DGJ and NB-DNJ treated mice testis showed abnormal spermatids (arrow) unlike elongated spermatids seen in the control(arrow) group. Scale bar—25 micron). (**b**) NB-DNJ treated mice epididymal sperm had abnormal acrosome at all three doses, whereas ND-DGJ treated mice showed an increasing number of sperm with abnormal acrosomes with increasing NB-DGJ dose (sperm head indicated by arrows). Scale bar—50 micron.

**Figure 4 ijms-21-00301-f004:**
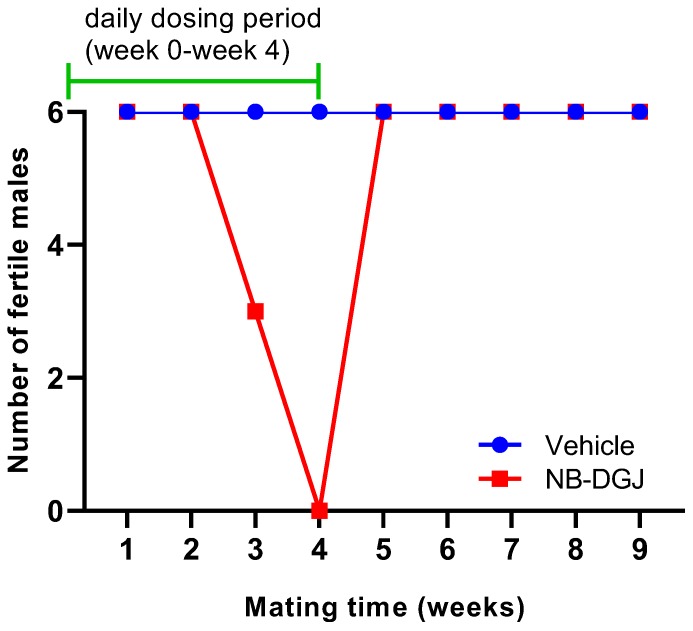
Effect of *N*B-DGJ on the fertility of male CD rats. The *x*-axis represents the week at which fertility was analyzed on the final day of the mating week and the y-axis denotes the number of rats (out of six) that were able to impregnate female rats. The green line indicates the weeks for which the rats were receiving daily treatment (weeks 0–4, days 0–34). All male rats were necropsied at week 11.

**Figure 5 ijms-21-00301-f005:**
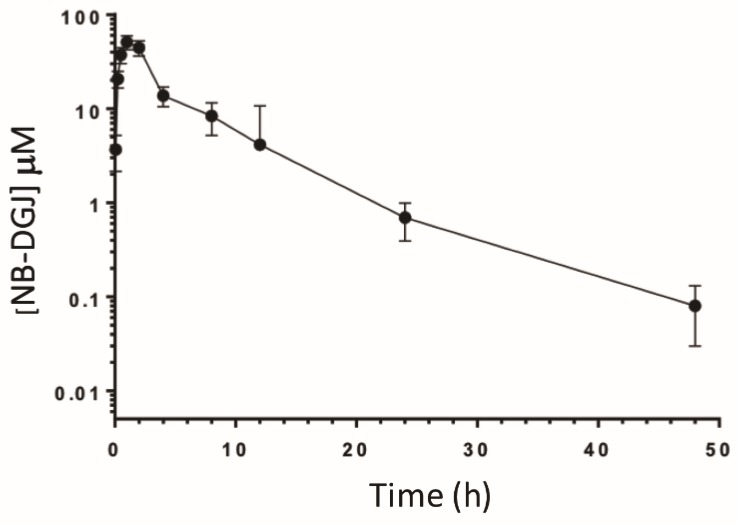
Pharmacokinetics of *N*B-DGJ in male CD rats. Rats (*n* = 4) with in-dwelling jugular vein cannulas were orally dosed with *N*B-DGJ (150 mg/kg) and at various times (0.083, 0.25, 0.5, 1, 2, 4, 8, 12, 24 and 48 h) afterwards, approximately 0.1 mL whole blood was collected from each animal. *N*B-DGJ was quantified in the resulting plasma using LC/MS/MS. Data are represented as the mean plasma levels ± S.D.

**Table 1 ijms-21-00301-t001:** Statistical analysis of normal and abnormal spermatozoa seen in mice treated with 50 mg/kg, 25 mg/kg and 15 mg/kg NB-DGJ. Numbers indicate the average percentage of spermatozoa population in all six mice/group.

Parameter	50 mg/kg		25 mg/kg		15 mg/kg	
SPERM MORPHOLOGY	Normal	Abnormal	Normal	Abnormal	Normal	Abnormal
AVERAGE (%)	11.78	88.21	37.85	62.05	48.4	51.59
STDEV	0.95	0.95	0.27	0.14	3.87	3.87

**Table 2 ijms-21-00301-t002:** Mean normal and abnormal implants.

Week of Study	Vehicle Conceptus	Fertile/Mated	*N*B-DGJ Conceptus	Fertile/Mated
1	14/0.3	6/6	14/0.3	6/6
2	13/0.7	6/6	6/0.1	6/6
3	13/0.8	6/6	1/0	3/6
4	13/0.5	6/6	0/0	0/6
5	14/0.2	6/6	10/2	6/6
6	13/0.3	6/6	13/0.3	6/6
7	14/0.2	6/6	13/0.3	6/6
8	13/0.3	6/6	14/0.3	6/6
9	13/0.6	6/6	13/0.5	6/6

Notes: Total numbers of conceptuses in both female rats housed per male were counted. Conceptuses = mean normal/mean abnormal conceptuses.

**Table 3 ijms-21-00301-t003:** Effect of single oral daily dose of *N*B-DGJ over 35 days on adult male CD rats. Parameters measured at necropsy (end of fertility trial; Week 11).

Treatment Group (*n* = 6)	Final Body Weight Week 11 (g ± SE)	Mean Paired Testis Weight (g ± SE)	Mean Spermatid Head Count/Testis # Cells × 10^6^	Mean Paired Epididymal Weight (g ± SE)	Mean Epididymal Sperm Motility (%)	Mean Epididymal Sperm with Normal Morphology (% ± SE)	Mean Seminal Vesicle Weight with Coagulating Gland (g)
vehicle control	563 ± 12	3.57 ± 0.10	122.32 ± 4.24	1.54 ± 0.07	87 ± 1	87 ± 1	1.58 ± 0.10
*N*B-DGJ ^a^	572 ± 17	3.80 ± 0.11	123.59 ± 3.73	1.64 ± 0.03	60 ± 3 *	72 ± 1 *	1.40 ± 0.11

Notes: ^a^
*N*B-DGJ at 150 mg/kg/day on days 0–34. * significant difference observed.

**Table 4 ijms-21-00301-t004:** Inhibition of testicular CGT and GBA2 derived from adult C57BL/6J mice and LE and CD rat testes by *N*B-DGJ.

IC_50_ CGT Inhibition (µM) ^a^			IC_50_ GBA2 Inhibition (µM) ^a^		
C57BL/6J mouse	LE rat	CD rat	C57BL/6J mouse	LE rat	CD rat
76.0 ± 2.0	42.0 ± 29.0	>1000	15.7 ± 3.1	13.8 ± 2.1	10.5 ± 1.6

Notes: **^a^** The IC_50_ values were determined as the mean ± SE average of three assays each.

**Table 5 ijms-21-00301-t005:** Pharmacokinetic (PK) analysis of NB-DGJ.

AUCINF_Obs (h * ng/mL)	C_max_ (ng/mL)	T_max_ (h)	t_½__Lambda_z (h)
55,600.0	11,016.7	1.3	6.5

Notes: Performed using WinNonLin using a non-compartmental model; * = multiplication

**Table 6 ijms-21-00301-t006:** IC_50_ data for enzyme inhibition by *N*B-DGJ from the literature.

CGT IC_50_ (µM)	Reference	GBA2 IC_50_ (µM)	Reference
10.0	[36]	0.30	[36]
41.4	[20]	5.3	[21]
		1–6	[15]

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
