# Peer review of "N-Butyldeoxygalactonojirimycin Induces Reversible Infertility in Male CD Rats"

_ijms, 2019, doi:10.3390/ijms21010301_

Round 1

Reviewer 1 Report

1) Does the fertile 3 males after 3 weeks of treatment with NB-DGJ(NB) show normal number of newborns? 

2)Table 2 shows no statistics for the 3 groups of 6 males that were tested.

3)The title of Table 3 describes 35 days of NB treatment, however in the text its written 11 weeks(line 165).

4)How NB affects sperm motility after 4 weeks of NB treatment?

5) What is the level of NB in the blood after 4 weeks of NB treatment and later on after one week without treatment when the infertility is reversed?

6) line 215 and in the abstract,the males were not treated for 35 days !! Four weeks are 28 days !

Reviewer 2 Report

Comments of the manuscript entitled “N-Butyldeoxygalactonojirimycin induces reversible infertility in male CD rats” by Vijayalaxmi Gupta, Sheri A. Hild, Sudhakar R. Jakkaraj, Erick J. Carlson, Henry L. Wong, Gunda I. Georg, and Joseph S. Tash.

The aim of the present study was to determine if N-butyldeoxygalactonojirimycin induces infertility in male CD rats and if this infertility is reversible. The authors found that NB-DGJ, administered orally induced reversible infertility in male CD rats.

The manuscript contains interesting data that could be of impact on the future of contraception if these findings will be reproduced in the future by a well-designed randomized clinical trial study in humans.

The manuscript is well organized and well written. However, there are some specific issues that need to be addressed before publication:

Figure 3 a and b. Please make 2 different figures. In the same figures, put control in a separate position and all the other treatments aligned:

Control 15mg NBDNJ 25mg NBDNJ 50mg NBDNJ
Nothing 15mg NBDGJ 15mg NBDGJ 15mg NBDGJ

Line 123-124. The phrase "dose dependent increase" is not supported with statistics or table values. Please rephrase the sentence.

Line 172. Again, the word "correlates" is not supported with statistics. Please rephrase the sentence. e.g. "in line with".

In figure S3. Correct the legend.

Table 3. Please clarify the measures. Are the values Final vs. baseline? End of the trial? I'm not sure.

Line 319. Please include the number of rats treated with NB-DGJ.

I suggest the creation of a new figure (supplemental) with a diagram of all the treatments, number of animals, measurements, etc.

At the end of the discussion part. Please include a new paragraph with the limitations of your study as well as the applicability and suggest future works.

Thanks,

Reviewer 3 Report

In this study, the authors showed that an iminosugar was species specific and induced reversible infertility in male rat (one week after treatment) compared to that in male mouse (three weeks after treatment).

Major points:

This study aimed at demonstrating the different effect of iminosugar in male rat reproductive function, the essential analysis of testis histology (such as HE-stain) and sperm morphology in rat treated by NB-DNJ or NB-DGJ should be produced.

The severe aspermatogenesis was detected in the mice after NB-DNJ treatment than that in NB-DGJ treatment (Figure 3A and 3B), so the spermatogenesis in NB-DNJ treated rats is important to be experimentally verified.

Additionally, it has been demonstrated that NB-DGJ caused dose dependent increase in the number of abnormal spermatozoa in mice, the possibility of high dose NB-DGJ (over 150mg/kg) or NB-DNJ might cause long period infertility in rat as similar to that in mice.

The explanations about experimental design and experiment groups are not simple to understand, the time schedule of treatment applied to the mice and rats should be added. The number of samples in each experiment also need to be indicated.

The results about testis histology and sperm morphology of mice treated with 150mg/kg NB-DNJ or NB-DGJ are necessary as positive control in this study (in Figure 3 and Table 1). Furthermore, the sperm morphology average of mice treated with NB-DNJ also need to be shown in Table 1.

Minor points:

Some explanation in Results should be descripted in “Materials and Methods”, such as the part of “2.1. Synthesis of iminosugars” (line 105-112), line 132-133, line 135-143.

The method to calculate the average of abnormal sperm in mice treated with NB-DNJ or NB-DGJ (in Table 1) should be provided explanation in “Materials and Methods”.

The figure of mice testis should be Figure 3a and the figure of epididymal sperm should be Figure 3b.

“Normal” and “Control” are mixed in article and should be unified.

Some grammars, format and abbreviations are not appropriate, please correct.

Round 2

Reviewer 1 Report

Yes, I accept.

Author Response

We appreciate your suggestion to describe the results more clearly and have worked on your suggestion.

Reviewer 3 Report

Major points:

This study aimed at demonstrating the different effect of iminosugar in male rat reproductive function, the analysis of testis histology and sperm morphology in rat treated by NB-DGJ are necessary. The study on the effect of NB-DNJ in mouse has been well demonstrated, and the new finding in this study is the reversible infertility in rat.

The author concluded that “the rapid return of fertility indicates that the major effect of NB-DGJ may be epididymal rather than testicular” in Abstract, the data of testis histology and sperm morphology in rat treated by NB-DGJ should be provided.

Author Response

The authors appreciate your comments.

Response to reviewer’s comments (Round -2) Reviewer # 3: This study aimed at demonstrating the different effect of iminosugar in male rat reproductive function, the analysis of testis histology and sperm morphology in rat treated by NB-DGJ are necessary. The study on the effect of NB-DNJ in mouse has been well demonstrated, and the new finding in this study is the reversible infertility in rat. The author concluded that “the rapid return of fertility indicates that the major effect of NB-DGJ may be epididymal rather than testicular” in Abstract, the data of testis histology and sperm morphology in rat treated by NB-DGJ should be provided.
Response. The authors thank you for your critical review and appreciate your suggestion. We have included two graphs in supplementary information (Pg. S10), showing epididymal sperm motility (Fig. 4A) and sperm morphology (Fig. 4B). These graphs indicate that there was significant difference in epididymal sperm motility and morphology between vehicle and NB-DGJ treated rats even at the end of study. This information was included in Table. 3.
We have also described in the Discussion (Line 231-234), how fertility returned within one week of termination of NB-DGJ administration, which is just a little shorter than the sperm epididymal transit time in rats (~8 days). Further, as indicated in Table.3, mean paired testis weight and mean spermatid head count was similar between vehicle and NB-DGJ treated rats. This clearly indicates that NB-DGJ did not affect the testis in these rats. Moreover, if the testis was affected, fertility would not return this quick. Additionally, hormone analysis in male rats indicated that there was no significant difference in the levels of testosterone, inhibin B and FSH levels between vehicle and NB-DGJ treated rats. If testis was impaired, there would have been alterations in these hormone levels. Since, we had convincing evidence that testis was not affected, we decided not to pursue histology. As mentioned in the manuscript, the rat mating studies were carried out at NIH as part of a contract. Due to high costs involved in storage of specimen, NIH disposed of the tissues, since it was not of clinical relevance. We hope that you will consider the testis weight, spermatid head count, epididymal sperm motility, hormone level and fertility reversal data, which support that NB-DGJ affected the epididymis and not the testis.

Round 3

Reviewer 3 Report

The conclusion of this study is showing NB-DNJ induces reversible infertility in male rats that is not related to C57BL/6 mice. This is the first report about the effect of NB-DNJ on rat reproduction, so the certainty results are required. The author should add the NB-DNJ treated rat epididymal sperm staining with the graph epididymal sperm morphology, but not only the graph.

The study on the effect of NB-DNJ in mouse has been well demonstrated, and the new finding in this study is the effect NB-DNJ on rat testicular histology.